# Genetic Analysis of HPIV3 That Emerged during the SARS-CoV-2 Pandemic in Gwangju, South Korea

**DOI:** 10.3390/v14071446

**Published:** 2022-06-30

**Authors:** Hongsu Lee, Sun-Hee Kim, Sun-Ju Cho, Yeong-Un Lee, Kwangho Lee, Yong-Pyo Lee, Jinjong Seo, Yoon-Seok Chung

**Affiliations:** 1Division of Emerging Infectious Disease, Department of Infectious Disease Research, Health and Environment Research Institute of Gwangju, Gwangju 61954, Korea; lhs9213@korea.kr (H.L.); sj0426@korea.kr (S.-J.C.); gloryw3@korea.kr (Y.-U.L.); twilight0930@korea.kr (K.L.); sjj21@korea.kr (J.S.); 2Division of Infectious Disease Diagnosis Control, Honam Regional Center for Disease Control and Prevention, Korea Diseases Control and Prevention Agency, Gwangju 61947, Korea; lyp0112@korea.kr

**Keywords:** human parainfluenza virus type 3, COVID-19, pandemic, surveillance

## Abstract

Community mitigation measures taken owing to the COVID-19 pandemic have caused a decrease in the number of respiratory viruses, including the human parainfluenza virus type 3 (HPIV3), and a delay in their occurrence. HPIV3 was rarely detected as a consequence of monitoring respiratory viral pathogens in Gwangju, Korea, in 2020; however, it resurfaced as a delayed outbreak and peaked in September–October 2021. To understand the genetic characteristics of the reemerging virus, antigenic gene sequences and evolutionary analyses of the hemagglutinin-neuraminidase (HN) and fusion (F) genes were performed for 129 HPIV3 pathogens prevalent in Gwangju from 2018 to 2021. Unlike the prevalence of various HPIV3 strains in 2018-2019, the prevalence of HPIV3 by strains with reduced diversity was confirmed in 2021. It could be inferred that this decrease in genetic diversity was due to the restriction of inflow from other regions at home and abroad following the community mitigation measures and the spread within the region. The HPIV3 that emerged in 2021 consisted of HN coding regions that were 100% consistent with the sequence identified in Saitama, Japan, in 2018, and F coding regions exhibiting 99.6% homology to a sequence identified in India in 2017, among the ranks reported to the National Center for Biotechnology Information. The emergence of a new lineage in a community can lead to a mass outbreak by collapsing the collective immunity of the existing acquired area; therefore, continuous monitoring is necessary.

## 1. Introduction

Human parainfluenza viruses (HPIVs) are an important cause of acute respiratory tract infections (ARIs), including lower respiratory tract infections, which are leading causes of morbidity and mortality in infants and young children [1,2]. HPIVs are a paraphyletic group of four distinct single-stranded RNA viruses that belong to the Paramyxoviridae family. They are classified into two genera based on their genome: HPIV1 and HPIV3 as genus Respirovirus, and HPIV2 and HPIV4 as genus Orthorubulavirus [3,4]. HPIV3 is the second most common cause of ARIs in infants and young children, after the human respiratory syncytial virus [5]. Epidemics of HPIV-3 infections usually occur in late spring and summer [6]. 

We have been participating in a nationwide surveillance network termed the Korea Influenza and Respiratory Virus Surveillance System (KINRESS) for monitoring ARIs in Korea. Throat or nasal swabs were collected from outpatients registered with ARIs throughout the year at regional collaborative hospitals. These samples were analyzed using real-time RT-PCR to test for respiratory viruses, including HPIVs. According to the KINRESS data from 2015 to 2019, HPIV3 has been recurring in Gwangju every year between March and September. 

Since the Wuhan government first reported 27 cases of pneumonia on December 31, 2019, its global impact on public health has been disastrous. On February 18, 2020, the Korean government identified a large-scale outbreak of severe acute respiratory syndrome coronavirus 2 (SARS-CoV-2) that mainly occurred in religious organizations; thus, mitigation measures such as rapid patient detection, patient isolation, contact tracking, and isolation began. Community mitigation measures to cope with COVID-19 also affect the circulation of other respiratory viruses, such as HPIVs [7,8]. In South Korea, HPIVs did not occur in 2020, but the occurrence of a delayed annual HPIV3 outbreak in 2021 was reported [9].

HPIV3 contains two major glycoproteins: hemagglutinin-neuraminidase (HN) and the fusion (F) protein. The HN protein plays a role in viral attachment to the host, invasion, isolation, and propagation, and the F protein is known to play an important role in membrane fusion for viral entry, and syncytium formation [10,11]. Therefore, the sequences of the HN and F genes are utilized to study the phylogenetic relationship between different HPIV-3 variants circulating globally [12].

Sequencing the HN and F genes in this study enabled us to illustrate the genetic characteristics of HPIV3 strains that reemerged after mitigation measures against the COVID-19 pandemic were taken.

## 2. Materials and Methods

### 2.1. Sample Collection

Throat or nasal swabs were collected from outpatients registered with ARIs at regional collaborative hospitals in the Gwangju area. HPIV3-positive samples were detected among samples from 2018 to 2021, before and after the COVID-19 pandemic.

### 2.2. Sequencing of HN and F Genes

From the samples identified as HPIV3 positive among the respiratory specimens collected via the KINRESS in the Gwangju area, 40 were randomly selected in 2018, 40 in 2019, and 49 in 2021. Viral RNA was extracted using a QIAamp Viral RNA Mini Kit (Qiagen, Hilden, Germany) according to the manufacturer’s instructions. RT- PCR was performed using the primer pair from Table 1 for the hemagglutinin-neuraminidase (HN; 450 nt) and fusion (F; 650 nt) amplicons, and PCR conditions were as follows: 30 min at 50 °C, followed by 5 min at 95 °C and 40 cycles of 30 s at 95 °C, 30 s at 50 °C, and 30 s at 72 °C, and a final elongation step at 72 °C for 10 min (Table 1) [13]. PCR products were subsequently sequenced using the BigDye Terminator v3.1 Cycle Sequencing Kit in an Applied Biosystems 3730XL DNA Analyzer (Applied Biosystems, San Francisco, CA, USA) in both directions.

### 2.3. Phylogenetic Analyses

The sequencing results of the HN and F genes in both directions were assembled into consensus sequences using the CLC Genomics Workbench (version 21.0.3). In the present study, phylogenetic analyses of the HN and F sequences were performed together using homologous nucleotide sequences from previous molecular epidemiological studies that were retrieved from the NCBI GenBank database as references (Table 2). Phylogenetic analyses were performed using the maximum likelihood method with the Tamura 3-parameter model in MEGA software (version 11.0.8., http://www.megasoftware.net, accessed on 1 October 2021) [14]. The reliability of the internal branch was assessed using non-parametric bootstrap analysis with 1000 replicates.

To illustrate the genetic distance between all strains, we calculated pairwise distances using the MEGA X software. The Tamura 3-parameter model was used as a substitution model using bootstrap analysis with 1000 replicates. Unlike the phylogenetic analyses, pairwise distances were calculated without references to clearly indicate year-wise differences.

## 3. Results

### 3.1. HPIV3 Epidemiology

In 2018 and 2019, according to KINRESS in the Gwangju area, the epidemic of HPIV3 peaked in May and showed the same seasonal pattern from March to August. Surprisingly, HPIV3 was barely detected in 2020, coinciding with the implementation of community mitigation measures against the COVID-19 pandemic. In 2021, HPIV3-positive cases collected by KINRESS in the Gwangju area reemerged. In addition, the epidemic lasted from August to December, and the rate of detection was higher than that in 2018 and 2019 (Figure 1). In the age-wise distribution, those under the age of 2 years were infected at a significantly higher rate than those in other age groups in both 2018 and 2019 (Table 3).

### 3.2. Phylogenetic Analyses 

We were able to obtain a total of 123 partial nucleotide sequences for the HN coding region in this study, and these were used to construct a phylogenetic tree together with the reference sequences using the maximum likelihood method. Clusters, subclusters, and lineages were identified using the Tamura 3-parameter model in MEGA X software and a genetic distance of 0.045 (cluster), 0.020 (subcluster), and 0.010 (lineage) were identified [15]. All 123 sequences were identified as Cluster 1, and Cluster 1 was further subdivided into three subclusters, from 1a to 1c. Unlike the sequences from 2018 and 2019 that are divided into subclusters, all sequences from 2021 were identified as Subcluster 1c. Moreover, Subcluster 1c was further divided into two lineages, and in particular, it was confirmed that all strains from 2021 were identified as 1c lineage 1 (Figure 2). In addition, we calculated the pairwise distance among sequences within the same year, and it was confirmed to be 0.012 ± 0.011 in 2018 and 0.013 ± 0.008 in 2019 and no distance among sequences was detected in 2021 (Figure 4a).

The phylogenetic tree of the F coding region was also constructed with 123 nucleotide sequences and aligned according to the same methods and criteria as for HN gene analysis. For the F gene, two clusters were identified. Cluster 1 was further subdivided into subclusters, 1a and 1b, with smaller subdivisions into lineages (Figure 3). All sequences in 2021 were identified as 1a lineage 1 in Subcluster 1a. On the other hand, those in 2018 and 2019 were distributed into clusters and subclusters. The pairwise distances were 0.013 ± 0.015 in 2018, 0.013 ± 0.007 in 2019, and 0.000 ± 0.001 in 2021 (Figure 4b).

## 4. Discussion

In February 2020, measures such as social distancing following the COVID-19 pandemic in Korea affected respiratory viral activity, including HPIV3. HPIV3 rarely occurred in 2020; the season was delayed in 2021 and the scale was much larger than before. Changes in respiratory virus epidemic patterns following the COVID-19 pandemic have been reported to show similar trends not only in Korea, but also in countries around the world [8,16].

Since the outbreak of COVID-19 in December 2019, it has been confirmed that quarantine measures such as wearing masks and distancing recommended by the WHO are very effective in preventing respiratory infections [17]. In addition, unlike occurrences prior to 2019, most respiratory viruses, except for non-enveloped viruses such as rhinovirus, adenovirus, and bocavirus, did not occur in Korea during the epidemic of SARS-CoV-2 [7,18]. In the summer of 2021, when the COVID-19 outbreak continued, HPIV3 reemergence was confirmed in Gwangju. We thought that genetic analysis of existing viruses and viruses that reemerged in 2021 would be important to identify genetic associations with HPIV3 viruses that were prevalent before 2019 and to determine whether the reemergence of HPIV3 in Gwangju was caused by local reservoirs.

We needed to confirm the genetic characteristics of HPIV3, which had been prevalent since COVID-19. Like other RNA viruses, HPIV3 is known to have the ability to obtain random point mutations throughout the genome in surface glycoproteins (HN and F), subject to selective pressure from human immune responses [19]. Therefore, we compared the partial sequencing results of HN and F genes with the HPIV3 strain prevalent in Gwangju between 2018 and 2021. As a result, it was confirmed that the strains that emerged in 2021 had genetic characteristics differentiated from those that were prevalent in 2018-2019. Their HN coding regions showed 100% identical sequence to the strain isolated from Saitama, Japan in 2021 (GenBank accession No. LC486648) [20] and their F coding regions showed the highest homology with the sequence of the strain (GenBank accession No. MH330335) identified in India in 2017, at 99.6% (Table 2). It was also confirmed that the strains in 2021 were significantly reduced in diversity compared with those in 2018-2019. In this regard, it could be estimated that the cases in 2018 and 2019 occurred through various routes, but in 2021, it could be inferred that there was local transmission through the limited domestic and foreign entry routes due to the COVID-19 pandemic.

The HPIV3 identified in this study is the emergence of a new lineage in Gwangju, which could collapse the herd immunity of naturally existing, acquired, and maintained regions and occur locally on a large scale. In this case, the mortality rate can increase if immunocompromised people and people sensitive to infections, such as the elderly become infected. Therefore, it is necessary to continuously analyze genetic antigenic mutations in the virus, as well as monitor the occurrence of the virus.

## Figures and Tables

**Figure 1 viruses-14-01446-f001:**
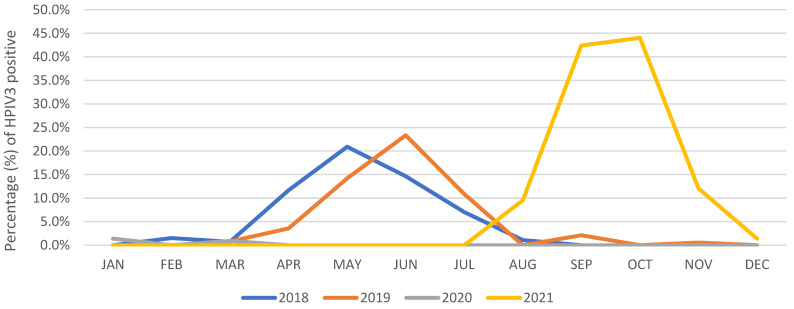
Seasonal patterns of the HPIV3 epidemic recorded in the Korea Influenza and Respiratory Virus Surveillance System (KINRESS) in Gwangju area between 2018 and 2021.

**Figure 2 viruses-14-01446-f002:**
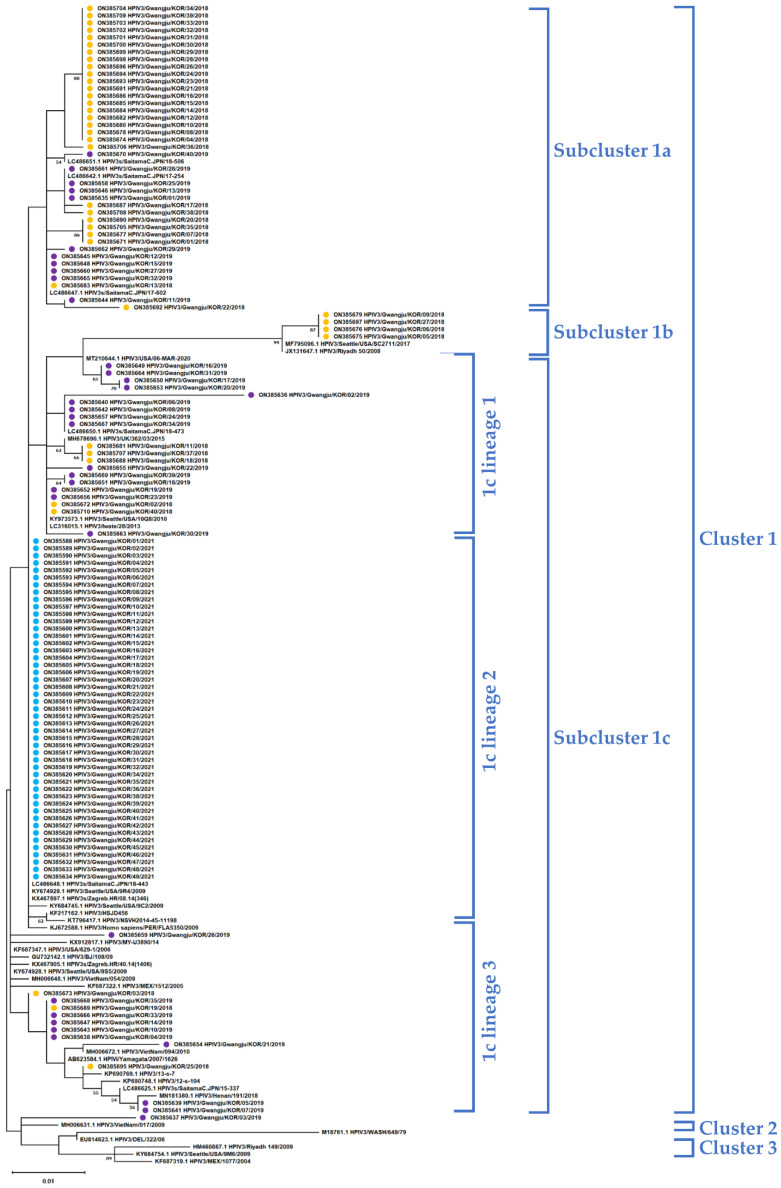
Phylogenetic trees of the hemagglutinin-neuraminidase (HN) gene for HPIV3 constructed using the maximum likelihood method. Phylogenetic analysis was performed using the maximum likelihood method with the Tamura 3-parameter model with 1000 bootstrap repetitions. Clusters, subclusters, and strains were identified using genetic distances calculated with MEGA X software (Clusters: 0.045; Subclusters: 0.02; Strains: 0.01). The GenBank accession numbers of all strains are indicated in parentheses.

**Figure 3 viruses-14-01446-f003:**
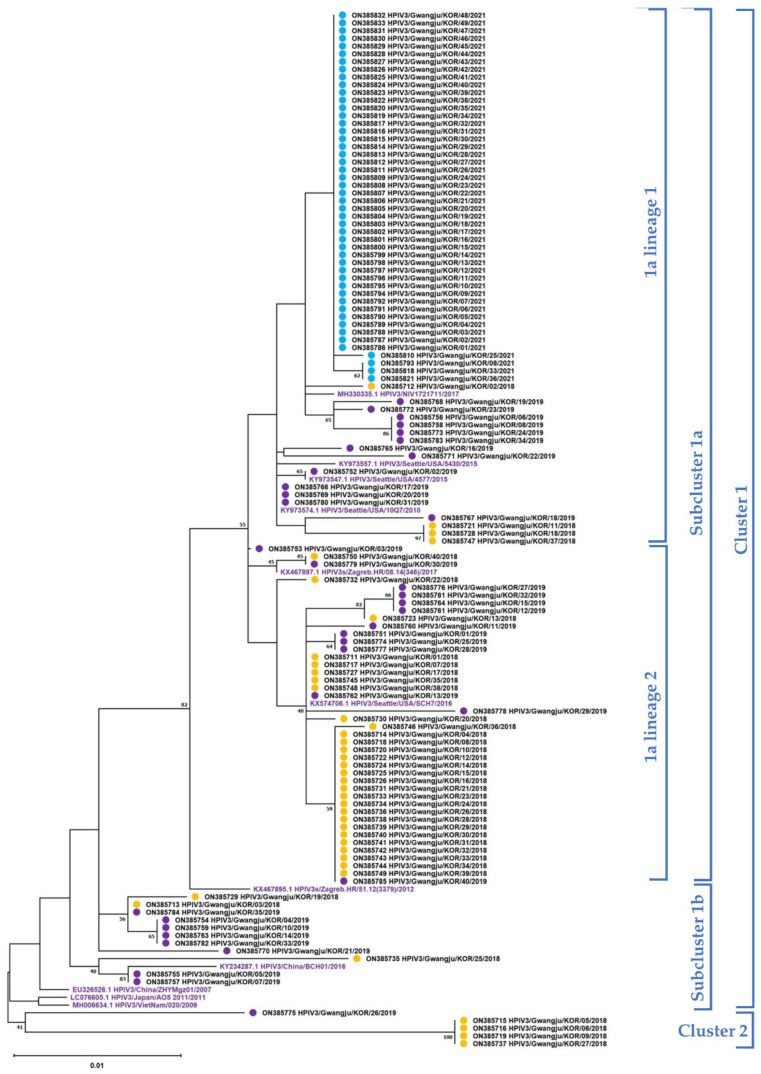
Phylogenetic trees of the fusion (F) gene for HPIV3 constructed using the maximum likelihood method. Phylogenetic analysis was performed using the maximum likelihood method with the Tamura 3-parameter model with 1000 bootstrap repetitions. Clusters, subclusters, and strains were identified using genetic distances calculated with MEGA X software (Clusters: 0.045; Subclusters: 0.02; Strains: 0.01). The GenBank accession numbers of all strains are indicated in parentheses.

**Figure 4 viruses-14-01446-f004:**
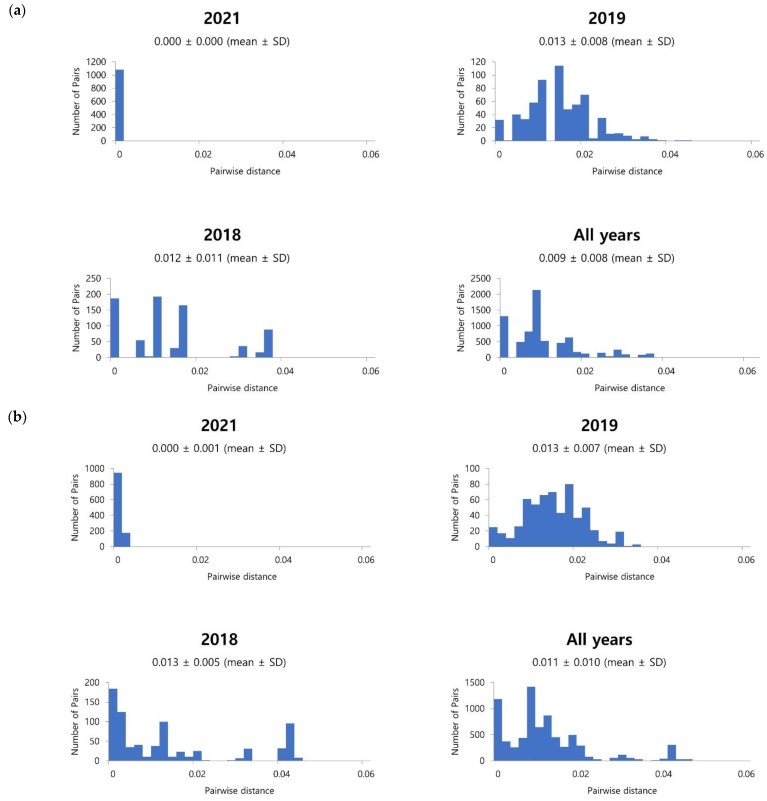
(**a**) Distribution of pairwise distance based on the nucleotide sequences of the HN gene; (**b**) Distribution of pairwise distance based on the nucleotide sequences of the F gene.

**Table 1 viruses-14-01446-t001:** Primers used for HN and F gene amplification.

Target	Oligonucleotide Sequence (5′-3′)		Size (bp)	Reference
HN	Forward	ATTACTCGAGGTTGCCAGGA	450	[13]
	Reverse	CCGCGACACCCAGTTGTG		
F	Forward	CTTTGGAGGGGTAATTGGAACTA	621	
	Reverse	ATGATGTGGCTGGGAAGAGG		

**Table 2 viruses-14-01446-t002:** List of HPIV3 reference strains included in phylogenetic analysis.

Strain	Accession No.	Country/Year
HPIV3s/SaitamaC.JPN/18-443	LC486648	Japan/2018
HPIV3s/SaitamaC.JPN/18-506	LC486651	Japan/2018
HPIV3s/SaitamaC.JPN/17-602	LC486647	Japan/2017
HPIV3s/SaitamaC.JPN/17-254	LC486642	Japan/2017
HSJD456	KF217162	Spain/2010
Henan/191/2018	MN181380	China/2018
HPIV3s/SaitamaC.JPN/15-337	LC486625	Japan/2015
HPIVi/Yamagata/2007/1626	AB623584	Japan/2007
HPIV3s/SaitamaC.JPN/18-473	LC486650	Japan/2018
HPIV3/Iwate/28/2013	LC316015	Japan/2013
HPIV3/UK/362/03/2015	MH678690	UK/2015
USA/06-MAR-2020	MT210644	USA/2020
HPIV3/Seattle/USA/SC2711/2017	MF795096	USA/2017
NIV1721711	MH330335	India/2017
HPIV3/Seattle/USA/5430/2015	KY973557	USA/2015
HPIV3s/Zagreb.HR/08.14(346)	KX467897	Croatia/2014
HPIV3/Seattle/USA/SCH7/2016	KX574706	USA/2016
HPIV3s/Zagreb.HR/51.12(3379)	KX467895	Croatia/2012
HPIV3/China/BCH01/2016	KY234287	China/2016
ZHYMgz01	EU326526	China/2007
AO5_2011	LC076605	Japan/2015
HPIV3/Vietnam/020/2009	MH006634	Vietnam/2009
HPIV3/DEL/322/06	EU814623	India/2006
HPIV3/DEL/w32/05	EU814625	India/2005
HPIV3/BJ/108/09	GU732142	China/2009
Riyadh 149/2009	HM460887	Saudi Arabia/2009
Riyadh 50/2008	JX131647	Saudi Arabia/2008
HPIV3/MEX/1077/2004	KF687319	Mexico/2004
HPIV3/MEX/1512/2005	KF687322	Mexico/2005
HPIV3/USA/629-1/2006	KF687347	USA/2006
HPIV3/Homo sapiens/PER/FLA5350/2009	KJ672588	Peru/2009
12-s-104	KP690748	China/2012
13-s-7	KP690769	China/2013
NSVH2014-45-11198	KT796417	Spain/2014
HPIV3s/Zagreb.HR/08.14(346)	KX467897	Croatia/2014
HPIV3s/Zagreb.HR/40.14(1406)	KX467905	Croatia/2014
MY-U3890/14	KX912817	Malaysia/2014
HPIV3/Seattle/USA/9S5/2009	KY674928	USA/2009
HPIV3/Seattle/USA/9R4/2009	KY674929	USA/2009
HPIV3/Seattle/USA/9C2/2009	KY684745	USA/2009
HPIV3/Seattle/USA/9M6/2009	KY684754	USA/2009
HPIV3/Seattle/USA/10Q8/2010	KY973573	USA/2010
WASH/1511/73	M18759	USA/1973
AUS/124854/74	M18760	USA/1974
WASH/649/79	M18761	USA/1979
HPIV3/Vietnam/017/2009	MH006631	Vietnam/2009
HPIV3/Vietnam/029/2009	MH006641	Vietnam/2009
HPIV3/Vietnam/054/2009	MH006648	Vietnam/2009
HPIV3/Vietnam/094/2010	MH006672	Vietnam/2010

**Table 3 viruses-14-01446-t003:** HPIV3 Epidemiology in Gwangju area, South Korea, from 2018 to 2021.

Year			2018	2019	2020	2021
Total no. of samples collected according to KINRESS			1433	1545	1286	1348
HPIV3-positive samples	Total No.		62	64	4	126
	Sex	Male	23	37	4	55
		Female	39	27		71
	Age group (years)	<2	1	4		38
		2 to 9	43	44	2	79
		10 to 19	5	6	1	1
		>19	13	10	1	8
	Month	JAN			3	
		FEB	2			
		MAR	1	1	1	
		APR	14	5		
		MAY	28	24		
		JUN	12	21		
		JUL	4	10		
		AUG	1			6
		SEP		2		53
		OCT				48
		NOV		1		17
		DEC				2

## Data Availability

The datasets generated for this study can be found in the online repositories. All sequences in this study are deposited in the Genbank database (accession numbers: ON385588-385833).

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
