# Peer review of "Genetic Analysis of HPIV3 That Emerged during the SARS-CoV-2 Pandemic in Gwangju, South Korea"

_viruses, 2022, doi:10.3390/v14071446_

Round 1

Reviewer 1 Report

In this interesting study, the authors probe the origins of the reemergent HPIV3 virus circulating in Gwangju, South Korea in the fall of 2021, after the region and the world had been under strict control measures in response to the ongoing Covid-19 pandemic.  As one would clearly expect, the measures introduced to curb the spread of Covid-19 also stemmed the spread of other respiratory pathogens, including HPIV3.  Thus, careful analyses of the newly emerged HPIV3 strains makes it possible to definitively identify the etiology of the 2021 virus(es).  Certainly it is clear that an understanding of the identification and etiology of the new virus is important to protecting especially vulnerable parts of the population.

To this end, the authors have sequenced the HN and F genes of the reemergent 2021 virus and compared these sequences to those of viruses circulating in the region prior to the onset of the pandemic.  Their data definitively show that the newly emerged 2021 isolates consist of HN coding regions most similar to the Saitama, Japan 2018 isolate and F coding regions consistent with sequences identified in India in 2017.

The data seem to show quite definitively that the new viruses possess an HN gene predominantly derived from the Japanese isolate and an F gene similarly predominantly derived from the Indian isolate.  I found the organization and presentation of the data very strong and convincing.  But, the question remains as to the etiology of the new viruses.  If the authors are concluding that the new virus possesses HN and F from divergent viruses, the reader can reasonably expect the authors to offer some speculation as to how such a virus might have arisen.

Author Response

Since only one file can be attached, the edited manuscript is attached to the response file.

Reviewer 2 Report

Lee et al., investigate the prevalence of PIV3 (an important human pathogen) over the course of the pandemic (before, during lockdown, and afterwards) by PCR-based methods and viral gene sequencing. They show a clear drop in detection during lockdown and then an increase afterwards. Sequencing shows that the viruses after lockdown are distinct from those before and that there appears to be reduced diversity. This is a timely and important area of research given uncertainties surrounding the behaviour of viruses after lockdowns and in the presence of SARS-CoV-2. The strengths of this work is the sample size and covered time period (before and after pandemic), and the use of 2 gene sequencing from across the same time period. However, I feel that there could be significant improvements to the phylogenetic analysis that would greatly strengthen the work given that in both trees there is very little support for any groups. I also have several minor comments.

Regarding phylogenetic methods, I found this preprint (https://www.biorxiv.org/content/10.1101/2022.03.15.484550v1.full.pdf+html)  on PIV3 detection and HN sequencing from S Korea that while has a more limited selection of samples and sequences, has more convincing phylogenetic analyses that could be used as a template to improve the current study. Without seeing the alignment from which the analyses are based in Lee et al., I am not sure how to precisely improve the analysis, but the authors could i) show the alignment for HN and F partial nucleotide sequences, try different substitution models in line with literature, and could consult this other recent PIV3 phylogenetics study: https://www.ncbi.nlm.nih.gov/pmc/articles/PMC6281019/ One possibility may be to include a human PIV1 outgroup.

Title: - provided phylogenetic analysis is improved, the title could include reference to observed reduced diversity in recent sequences.

Abstract

Careful with terminology: I would not say a novel virus appeared, I would say a novel lineage or undetected lineage. Also Can't say it was due to importation as could be due to limited sampling within country

Intro:

Update taxonomy - PIV2 and 4 are members of the Orthorubulavirus genus https://talk.ictvonline.org/ictv-reports/ictv_online_report/negative-sense-rna-viruses/w/paramyxoviridae/1190/genus-orthorubulavirus

Major glycoproteins (both proteins are glycosylated) - line 54

F is essential for entry and fusion not just in syncytium-formation

Methods:

More info on samples needed - how were they taken precisely? Nasal? Saliva?

PCR conditions needed

Is the phylogenetic tree made using nucleotide or amino acid sequences?

Results

Would sample % positivity better as possibly more samples are being taken and tested?

Can you identify novel-lineage specific amino acid changes of possible relevance to infection?

Discussion

Epidermal glycoproteins - not sure what this means line 157

Hn and f should give similar patterns as no recombination - is this what you observe?

Can't say it was due to importation as could be due to limited sampling within country

References

There seems to be fairly limited number of references used.

Author Response

(The authors gave the same response as above.)

Round 2

Reviewer 2 Report

The authors have addressed all my concerns.